# Mesoporous Carbon: A Versatile Material for Scientific Applications

**DOI:** 10.3390/ijms22094498

**Published:** 2021-04-26

**Authors:** Md. Motiar Rahman, Mst Gulshan Ara, Mohammad Abdul Alim, Md. Sahab Uddin, Agnieszka Najda, Ghadeer M. Albadrani, Amany A. Sayed, Shaker A. Mousa, Mohamed M. Abdel-Daim

**Affiliations:** 1Shenzhen Institute of Advanced Technology (SIAT) of the Chinese Academy of Sciences (CAS), Shenzhen 518055, China; 2Nanotechnology and Catalysis Research Center (NanoCat), University of Malaya, Kuala Lumpur 50603, Malaysia; aramstgulshan63@gmail.com; 3Department of Biochemistry and Molecular Biology, University of Rajshahi, Rajshahi 6205, Bangladesh; 4Department of Chemistry, Bangabandhu Sheikh Mujibur Rahman Science and Technology University, Gopalganj 8100, Bangladesh; alimbsmrstu@gmail.com; 5Graduate School of Innovative Life Science, University of Toyama, Gofuku 3190, Toyama 930-8555, Japan; 6Department of Pharmacy, Southeast University, Dhaka 1213, Bangladesh; msu-neuropharma@hotmail.com; 7Pharmakon Neuroscience Research Network, Dhaka 1207, Bangladesh; 8Laboratory of Quality of Vegetables and Medicinal Plants, Department of Vegetable Crops and Medicinal Plants, University of Life Sciences in Lublin, 15 Akademicka Street, 20-950 Lublin, Poland; agnieszka.najda@up.lublin.pl; 9Department of Biology, College of Science, Princess Nourah bint Abdulrahman University, Riyadh 11474, Saudi Arabia; gmalbadrani@pnu.edu.sa; 10Zoology Department, Faculty of Science, Cairo University, Giza 12613, Egypt; amanyasayed@sci.cu.edu.eg; 11Pharmaceutical Research Institute, Albany College of Pharmacy and Health Sciences, Rensselaer, NY 12144, USA; shaker.mousa@acphs.edu; 12Pharmacology Department, Faculty of Veterinary Medicine, Suez Canal University, Ismailia 41522, Egypt; abdeldaim.m@vet.suez.edu.eg

**Keywords:** mesoporous carbon, surface modification, catalytic support, adsorbent, drug delivery, capacitor

## Abstract

Mesoporous carbon is a promising material having multiple applications. It can act as a catalytic support and can be used in energy storage devices. Moreover, mesoporous carbon controls body’s oral drug delivery system and adsorb poisonous metal from water and various other molecules from an aqueous solution. The accuracy and improved activity of the carbon materials depend on some parameters. The recent breakthrough in the synthesis of mesoporous carbon, with high surface area, large pore-volume, and good thermostability, improves its activity manifold in performing functions. Considering the promising application of mesoporous carbon, it should be broadly illustrated in the literature. This review summarizes the potential application of mesoporous carbon in many scientific disciplines. Moreover, the outlook for further improvement of mesoporous carbon has been demonstrated in detail. Hopefully, it would act as a reference guidebook for researchers about the putative application of mesoporous carbon in multidimensional fields.

## 1. Introduction

According to IUPAC, the term “mesoporous carbon” refers to solid-based material, which might have either disordered or ordered networks with narrow or broad pores distribution in the range from 2 to 50 nm [1]. Historically, Ryoo et al. [2] reported the first structural composition of highly ordered mesoporous carbon material using mesoporous silica template, and later, much consideration was given to its synthetic methods and applications [3,4,5]. Therefore, rapid development was started for the synthesis of mesoporous carbon with the use of carbon precursors and self-assembly of copolymer molecular arrays [6,7,8,9]. Recently, modification, as well as the introduction of physicochemical properties on carbon, has been applied by incorporating inorganic constituents either anchored onto the mesoporous walls or decorated within the channels. Thus, the emergence of so-called modified mesoporous carbon has mostly driven the advancement of carbon-based electrodes or catalysts for its putative applications [10,11,12,13]. On the other hand, the substantial development of clean energy technology has motivated carbon-based researchers to exploit this porous material in the formation of energy storage devices [14,15,16,17,18].

Porous carbon materials, including carbon molecular sieves and activated carbon, have been synthesized on a large scale from fruit shells, wood, coal, or polymers by simple pyrolysis and physical or chemical treatment at elevated temperatures [19,20,21,22,23]. Moreover, these carbons have relatively broad pore-size distributions in both mesopore and micropore ranges and have been synthesized from defects resulting from heteroatoms elimination during carbonization. However, the carbon materials produced in these methods provide poor conductivity, structural integrity, and mass transport, owing to the presence of heteroatoms, lack of structural control, and restricted flow pathways. These limitations have been resolved by the introduction of mesopores for potential applications [24,25]. Mesoporous carbons with good physical properties are being synthesized on a large scale by carbonization from organic sources, such as sucrose, under acid treatment using silica templates. In this protocol, a transient silica-carbon complex is formed, and finally, following the alkali treatment, the silica is removed, leaving mesoporous carbon with analogous properties of silica materials (Figure 5) [24,25]. The mesoporous carbon synthesis, in this way, possesses good structural features and meets high physiological demands in modern scientific research [17,21,22,23].

Carbon materials with well-ordered mesoporous structures are ubiquitous and indispensable in modern scientific research, which attracted huge attention for their promising applications in various fields, including adsorption and separation of large biomolecules, electrical double-layer capacitors, catalytic support, wastewater treatment, and air purification (Figure 1) [26,27,28,29,30,31,32]. The emergence of inorganic porous materials, such as silica, zeolite, zeotype materials, carbon, metal oxides, and composites, has paved the way for the progression of drug delivery systems and other applications due to their high adsorption efficiency and ideal biocompatibility [33,34,35,36,37,38,39]. Compared with conventional porous materials, an inorganic carrier with small particle sizes, well-control shape, high stability, surface functionalization properties, low density, and wide availability offers substantial benefits in terms of application [40,41]. Recent advances in nanotechnology have provided a new dimension in carbon structure through continuous development of conventional methods as well as the introduction of new synthesis techniques [42]. Considering the well-controlled structure and promising application of mesoporous carbon in many scientific disciplines, it should be broadly compiled in the scientific literature from different points of view. To this line, the article aimed to provide a brief but elaborate overview of the applications of mesoporous carbon in many scientific research fields with special emphasis on catalytic reaction, adsorption, and drug delivery. Moreover, the outlook for further improvements and plausible challenges of mesoporous carbon has been demonstrated in detail. The authors, therefore, hope that this article would serve as a reference guidebook for future researchers regarding the potential application of mesoporous carbon in diverse research fields

## 2. Versatility of Mesoporous Carbon

Porous carbon nanomaterials are of great interest in current research due to their physicochemical properties and high surface area. However, traditional methods produce indiscriminate porous carbon materials with little to no control over their mesostructures and pore-size distributions, etc. The latest advances in the synthesis of other porous nanomaterials have led to developing alternative syntheses of mesoporous carbon with exceptionally ordered mesostructures and high surface areas, which play a pivotal role in separating media, cutting-edge electronic materials, and catalytic support (Figure 1, Table 1) [20,43]. Here, we outlined versatile applications of mesoporous carbon with the current scenarios in the respective fields.

### 2.1. Catalytic Supports

Currently, mesoporous carbon materials have drawn significant attention in the field of catalysis due to their promising structural features, such as unique optoelectronic and physio-chemical properties, uniform and tunable pore size, high surface area (up to 2500 m^2^/g), ordered pore architecture, good electrical conductivity and large pore volume, thermal stability, high recyclability, and biocompatibility [25,92,93]. These features offer high dispersion of metal nanoparticles, and therefore, increase the transport of ions, or electrons, and molecules, via the nanochannels/nanopores, and enhance product yields of a catalytic reaction with the shortest duration and minimum cost.

Delidovich et al. [44] comparatively analyzed the aerobic oxidation of glucose over Au/Al_2_O_3_ and Au/C catalysts at various glucose to Au molar ratios. The analysis showed that the Au/C catalysts were lower active than alumina-supported catalysts at high glucose to Au ratios. At these ratios, Au/Al_2_O_3_ (with various metal dispersion) exhibited high TOF (turnover frequency), which is a feature of the alumina-supported catalysts with metal particles size 1–5 nm. At these conditions, more than 90% of Au/Al_2_O_3_ catalysts showed a homogeneous gold distribution into the solid supports (Figure 2a). For Au/C catalysts with a non-homogeneous gold distribution, the superficial reaction rate was disturbed by internal diffusion, although the interface of gas–liquid–solid oxygen transfer influenced the overall reaction kinetics as well. In the aqueous phase, the reaction rate was controlled by oxygen dissolution at low glucose to Au ratios. In this condition, the carbon-supported catalysts were more active compared to the alumina-supported gold catalysts because the hydrophobic carbon supports highly adhere to the gas–liquid interface, and thus, facilitate the oxygen mass transfer towards catalytic sites (Figure 2b).

Ma et al. [94] have reported that although Au/C catalysts showed potential activity for specific liquid-phase oxidation; however, they are poorly active for gas-phase CO oxidation. This might be owing to several factors: first, carbon supports are acidic with low isoelectric points; second, during catalyst syntheses, carbon materials (acting as reducing agents) might interreact with Au^3+^ to form large/bulky gold nanoparticles [95]. In order to activate catalysts, carbon supports were modified in aqueous KMnO_4_ [51]. In aqueous solution, mesoporous carbon acts as a sacrificial reductant undergoing a self-limiting reaction for the decoration of MnO_x_ on its surface, and therefore, facilitates gold loading onto MnO_x_/C support through deposition-precipitation [96]. This modified (Au/MnO*_x_*/C) catalyst showed higher activity for CO oxidation (Figure 2c) [51]. The surface modification of carbon support has also been performed using various metal oxides (ZnO, FeO*_x_,* TiO_2_, etc.) for supported gold catalysts preparation [52,56,97].

Ketchie et al. [95] comparatively studied carbon-supported gold (5–42 nm) and unsupported gold powder in the aqueous-phase oxidation of glycerol and carbon monoxide (CO). For aqueous-phase oxidation of CO at 27 °C and pH = 14, supported-gold nanoparticles (5 nm) showed a TOF of 5 s^−1^, whereas large (42 nm) and bulky gold particles showed a TOF of only 0.5 and 0.4 s^−1^, respectively. Moreover, the observed rate of peroxide production during CO oxidation was higher over small gold particles. Small-sized gold particles showed similar effects in aqueous phase glycerol oxidation (60 °C and pH 13.8), in which 5 nm gold nanoparticles showed TOF 17 s^−1^, and larger and bulky golds showed an order of magnitude lower activity. Surprisingly, larger-sized gold (>20 nm) nanoparticles exhibited higher selectivity to glyceric acid formation. The poor selectivity of small Au particles has been attributed to the high rate of H_2_O_2_ formation during glycerol oxidation, since peroxide production promotes C-C bond cleavage. Prati et al. [98] prepared gold nanoparticles on various support materials by different methods for the liquid-phase oxidation of diols. Under normal conditions, the activity of Au/C catalysts showed different conversion and selectivity for ethane-1, 2 diol oxidations to glycolate.

### 2.2. Adsorbent

Nowadays, the carbons with well-ordered mesopores provide crucial applications in diverse fields, and one such example is the adsorption and separation of biomolecules, including enzymes, sugars, vitamins, proteins, amino acids, and useful antibiotics [42,99,100,101,102]. The carbon materials have good physicochemical properties with tunable pore diameter, high pore volume, and high surface area; however, the hydrophobicity and inertness of mesoporous carbons might not be favorable for this application. Therefore, surface modification of the carbon materials is required for their development and to change the hydrophobic and hydrophilic properties of the carbon surface [103]. Functional groups, such as the carboxyl group, are generally introduced into carbon surfaces upon oxidation using various oxidizing agents, for example, ammonium persulfate solution (APS), concentrated nitric acid, ozone, and sulfuric acids, and thus, enhances the wettability of polar solvents and activates the carbon surface to covalently immobilize DNA, nanocrystals, proteins, metal-containing complexes, etc. The extent of biomolecules adsorption on the carbon surface predominantly depends on the strength and density of the functional agent of the solid surface. Contemporary studies have shown that porous carbon treatment with APS provides more carboxyl groups onto the carbon surface compared to the treatment with H_2_SO_4_ or HNO_3_ because treatments with these strong acids offer large quantities of oxygen interfaces that affect the overall structure as well as textural properties [57,104].

The Vinu group [57] comparatively analyzed the ability of lysozyme adsorption using various APS-treated functionalized carbons namely CMK-3–0 h, CMK-3–12 h, CMK-3–24 h, and CMK-3–48 h (depending on the APS concentrations and oxidation times). However, these materials did not show promising activity for the adsorption of large biomolecules, owing to their poor textural features, disordered porous structure, and small pore size. The carboxyl group functionalization onto APS-treated-mesoporous carbon provides higher adsorption ability because the adsorbent molecules easily access the interior portion of mesoporous carbon materials and the oxidation of small carbon rods that are usually organized either vertically or horizontally inside the functionalized mesoporous channels. It has also been noticed that the anchoring capacity of carboxyl groups placed at the entrance and inside the mesopores may hinder the release of protein biomolecules from porous channels of carboxy functionalized mesoporous carbon, which results in elevated protein adsorption [57,105].

It has also been found that the lysozyme adsorption isotherms of carboxy group functionalization on APS-treated mesoporous carbon exhibited a sharp initial rise, advocating an extraordinary affinity between the carbon adsorbent and lysozyme. Among the carboxyl-functionalized-APS-treated materials, surprisingly, CMK-3–12 h and CMK-3–24 h exhibited higher adsorption abilities than untreated mesoporous material and CMK-3–48 h (Figure 3a). Moreover, the amount of protein adsorption depends on solution pH, which increases with increasing pH (Figure 3b) [57]. 

Hartmann et al. [58] demonstrated the adsorption ability of vitamin E onto different mesoporous carbon (CMK-1 and CMK-3) materials from two different solvents, n-butanol and n-heptane. The adsorption efficacy of these mesoporous materials was also compared with conventional adsorbents, including activated carbon and microporous carbon. The extent of vitamin E adsorption was found to largely rely on solvent types, as well as surface area and mesopore volume of the adsorbent. Moreover, it has been demonstrated that nonpolar n-heptane is preferred for higher vitamin E adsorption onto solid surfaces than polar solvent n-butanol (Figure 3c). This reduced adsorption ability of carbon materials in n-butanol apparently due to the increased production of n-butanol solvent clusters that may interact with ether and hydroxyl groups of the vitamin E. On the other hand, the interaction between the hydrophobic tail of vitamin E and the surface of hydrophobic mesoporous carbon was not influenced by the nonpolar n-heptane. In this report, CMK-3 (5.94 mmol/g) showed higher efficiency than CMK-1 (5.01 mmol/g) and activated carbon (4.10 mmol/g) in terms of vitamin E adsorption. XRD and N_2_ adsorption analyses showed that the vitamin E was closely packed inside mesopores of CMK-1 and CMK-3. This group also suggested that nonpolar solvent *n*-heptane is a promising candidate for higher loading of adsorbents with vitamin E. However, although further analysis is desired, when carbons would be loaded in medical applications with vitamin E, ethanol and water might be useful as solvents. 

Abe et al. [59] studied the adsorption ability of monosaccharides and disaccharides onto activated carbon from the complex aqueous solution. The adsorption features were demonstrated by the physical characteristics of adsorbates, e.g., molecular refraction and/or parachor. Moon and Cho [63] demonstrated that the activated carbon materials can act as the effective adsorbent for the separation and purification of maltopentaose from the complex mixture of other maltooligosaccharides, including maltotriose, maltose, and maltotetraose. Three distinct carbon materials were used for the purification of maltopentaose followed by enzymatic degradation. They employed activated carbon into the column and found 61% selectivity of maltopentaose by increasing the adsorption cycle. Lee et al. [60] analyzed the adsorption kinetics of monosaccharides (galactose, glucose, and fructose), disaccharides (sucrose, maltose, and lactose), and maltooligosaccharides (maltotriose, maltotetraose, and maltopentaose) onto 35 nm of mesoporous carbon. Among the nine saccharides used in this experiment, the adsorption of maltose was superior compared to other adsorbates at 100 °C. It was also found that the adsorption level of disaccharides, compared to monosaccharides, was extremely dependent on the chemical structure.

Antibiotics have extensive use in biomedical applications for disease inhibition and treatment, in aquaculture and in farming for growth promotion [106]. Owing to their unhealthy management, antibiotics are posing a potential hazard to ecosystems and public health [107]. Currently, a number of antibiotics are repeatedly measured in surface water, municipal wastewater, drinking water, and groundwater [108,109]. Among the exclusion methods for antibiotics from wastewater, adsorption is regarded as one of the most encouraging techniques due to its extraordinary removal efficacy, ecofriendly properties, and ease of adsorbent synthesis [110]. Activated carbon is regarded as a promising candidate for this application.

Ji et al. [61] prepared ordered micro-and mesoporous carbon materials to comparatively analyze the adsorption properties towards three antibiotics, namely tylosin, sulfamethoxazole, and tetracycline. Nonporous graphite, two commercial microporous activated carbons, and single-walled carbon nanotubes were also included for comparative analyses. The adsorption efficiency of smaller-sized sulfamethoxazole was higher onto activated carbons compared to other carbonaceous materials, which may be the result of the pore-filling effect. On the other hand, because of the size-exclusion effect, the adsorption of bulky tylosin and tetracycline was significantly lower over activated carbons, particularly for the more microporous absorbent, compared to the synthesized carbons. Following the normalization of the surface area of the adsorbent materials, adsorption of tylosin and tetracycline over prepared carbons was comparable with nonporous graphite, demonstrating the accessibility of the adsorbent surface area in adsorption. Compared to other porous adsorbents, the template synthesized carbons exhibited faster adsorption of tylosin and tetracycline, which may be due to their ordered-shaped, accessible as well as interconnected 3D pore structure. These results suggest that meso-and microporous carbons synthesized (template synthesis) in this study were promising candidates for the separation of antibiotics from an aqueous mixture.

Wang et al. [62] investigated the adsorption ability of three commonly used antibiotics, including sulfadiazine (SDZ), ciprofloxacin (CIP), and tetracycline (TC), from the aqueous phase over glucose-based mesoporous carbon (GMC). The highest adsorption efficiency of GMC for SDZ, CIP, and TC were 246.73, 369.85, and 297.91 mg/g, respectively. They also showed that the efficiency of antibiotics adsorption depends significantly on pH and the ideal pH values for the release of SDZ, CIP, and TC were measured as 4, 6, and 6, respectively. Antibiotic adsorption was not reported to be affected by ionic conditions, including NaCl, MgCl_2_, and CaCl_2_; however, the adsorption increased with an increasing temperature between 288 and 308 K of target antibiotics. The thermodynamic analyses showed that the adsorption process of selected antibiotics onto GMC was regarded spontaneous as well as exothermic in nature. The research proposed that several mechanisms, e.g., π-π interactions or hydrogen bonding, electrostatic interactions, and a hydrophobic effect, might be involved in the adsorption of antibiotics (Figure 4). Apart from these, several other reports on antibiotic adsorption have been illustrated in the literature [111,112,113,114,115].

### 2.3. Waste-Water Treatment

As per the Global Risk Report of 2015, the water catastrophe is positioned as the highest long-durable risk [116]. Approximately 783 million people have no access to safe and clean water, and 1 in 9 (11%) of the world’s population do not have access to drinking water [117]. Since population, as well as industry, are growing constantly, various pollutants, including endocrine disrupters, nitrosoamines, and heavy metals, are released into the water. These impurities have an adverse effect on environmental flora and fauna [118,119,120,121,122]. Currently, the conventional approaches of wastewater treatment, including sedimentation, coagulation, membrane process, decontamination, disinfection, ion exchange, and filtration, have been reported; however, these methods are intensive chemically and operationally [123,124,125,126,127,128,129,130,131,132,133]. Moreover, various water disinfectants have been applied for the chemical treatment of water, which may generate side products, for example, hydrochloric acid, chlorine, alum, ammonia, ozone, permanganate, and ferric salts, that are dangerous to freshwater resources [119]. Hence, there is an urgent need for efficient, large-scale, and low-cost systems for developing water quality. Many adsorbents have also been tested for metal removal, such as granular ferric hydroxide (GFH) [134,135], activated alumina [136,137], iron oxide-coated sand (IOCS) [138,139,140,141], and ferric (hydro)oxides [142,143]. Nowadays, ordered mesoporous carbons have drawn much attraction for this purpose due to their promising physical properties [144].

Gu et al. [65] impregnated iron onto mesoporous carbon, which diffuses ferrous (II) iron into the interior of porous carbon, so that the iron can disperse more consistently in the mesoporous carbon for improved arsenic adsorption (As^III^ and As^V^) from the drinking water (Figure 5). This mesoporous material showed a maximum of 5.15 mg and 5.96 mg As/g adsorption for arsenate and arsenite, respectively. Zhou et al. [66] synthesized iron-containing graphitic mesoporous carbon by carbonization using silica template and sucrose via impregnation. The synthesized complexes were shown high and rapid adsorption efficiency of pigments from *Carthamus tinctorius* flowers through solid-liquid magnetic separation. In a copious study, Zhang et al. [67] encapsulated mesoporous carbon with Fe_3_O_4_ for wastewater treatment. TEM analysis showed the complexes possess a rattle-like shape and Fe_3_O_4_ nanoparticles have reached the internal site of the mesoporous carbon. The porous wall of carbon materials provides enough spaces that enhance adsorption abilities as well as rates of pollutants adsorption from the aqueous solution. The synthesized composites show superparamagnetic properties with a saturation magnetization of 5.5 emu g^−1^ (electromagnetic unit per gram) at room temperature, which is the precondition for the high-magnetic separation of pollutants in wastewater treatment. The prepared samples showed higher organic pollutants adsorption when compared with commercially available activated carbon, and their highest adsorption ability for phenol, Congo red, and methylene blue reached 108.38, 1656.9, and 608.04, mg g^−1^, respectively.

Recently, novel magnetic mesoporous activated carbon (MMAC) was synthesized from activated carbon, isolated from rice husk via ZnCl_2_ chemical activation [64]. Mesostructural properties were induced through silica formation and the magnetic component was integrated by magnetite via wetness impregnation. This silica increases the porosity and gives a 3D network to stimulate the molecule adsorption. The adsorption ability of methyl orange and methyl blue dyes on the activated carbon, mesoporous activated carbon, and magnetic mesoporous carbon complex was studied. The effect of various parameters, for example, dosing of adsorbents, initial dye concentration, and pH of the dye solution, on the adsorption efficacy were analyzed. The MMAC adsorbed 98.5% methyl orange and 82% methyl blue dye in 30 min. The adsorbent material was recycled four times for methyl orange and methyl blue removal without a substantial loss in the adsorption ability. Comparative analyses showed that MMAC was more efficient in adsorbing wastewater than mesoporous activated carbon and activated carbon because of its high pore volume and high surface area. The magnetic properties of MMAC separate the adsorbent material after the adsorption process, showing that it is a promising material for water purification. Various other data on wastewater treatment using mesoporous carbon materials have also been reported in the recently published articles for the adsorption of malachite green dyes [68,69], methylene blue and methyl orange [70,71], hexavalent chromium [72,73,74,75,76,77,78,79,80,81,82], Cd(II) [73], copper [74], lead(II) [75,76,77], AV90 dye [78], pesticides [145], phenolic compound [79,80,81], diuron (herbicides), etc. [81].

### 2.4. Drug Delivery

Among various drug delivery routes, oral drug administration is still considered the preferred route because of patient compliance and simplicity [146,147,148]. The oral sustained-release formulation is the most common process and has drawn huge attraction in the disciplines of novel drug delivery. However, the oral delivery method is highly challenging because of its poor bioavailability as a result of biopharmaceutical (e.g., poor solubility and permeability and/or instability in the gastrointestinal environment) and pharmacokinetic (rapid clearance) challenges in the delivery process [149]. Until now, there is a surprisingly significant number of active drug constituents that show poor dissolution rates. For example, 30–40% of the leading 200 oral drug stuffs from the USA, Japan, Spain, and the UK are weakly dissolved in water [150]. Therefore, this is an emergency issue to explore novel drug formulation methods that might increase the water solubility of such drugs, predominantly the Biopharmaceutics Classification System (BCS) Class 2 compounds (poor solubility and high permeability) [151].

To this line, many delivery techniques, such as silica-lipid hybrid microcapsules [152], micelles [153], nanodispersions [154], silica nanoparticles [155,156,157], solid-lipid nanoparticles [158], self-microemulsification [159], and mesoporous carbon [43,160,161], have been established to upgrade drug bioavailability via improving the solubility and dissolution rate. There are several reports of drug delivery using mesoporous carbon with novel characteristics, e.g., higher drug loading, high adsorption efficiency, larger specific surface area and pore volume, and chemically inert [43,162]. In a recent study, mesoporous carbon with pore size (9.74 nm) and pore volume (1.53 cm^3^/g) was prepared for the delivery of celecoxib [83]. The drug was loaded into porous channels of carbon material. The performance was tested in terms of celecoxib release. The results showed that the rate of celecoxib release from mesoporous carbon was significantly higher than that of pure celecoxib administration. Mesoporous carbon increased the inhibitory function of celecoxib on the migration and invasion of human breast cancer cell lines (MDAMB-231 cells) that might be owing to the increased dissolution efficiency. It has been expected that the mesoporous carbon may have potential in the field of anticancer metastasis [83]. Wang et al. [162] selected a model drug (ibuprofen) and loaded it onto ordered mesoporous carbons to study the effect of carbon materials on drug release. They applied a two-step release process and observed a first rapid release, and subsequently, a slower release, and the delivery of ibuprofen was correlated with time. Moreover, the pore size was regarded as a vital factor for ibuprofen delivery, which stimulated positively with an increasing pore size. Miguel et al. [84] functionalized mesoporous carbon materials with a pH-sensitive self-immolative polymer and examined them as drug nanocarriers. A vial release assay of a Ru dye at pH 5 and 7.4 indicates the pH-sensitivity of the hybrid methods, exhibiting that only small quantities of the payload are released at neutral pH, while at acidic pH, self-immolation occurs and a substantial extent of the cargo is liberated (Figure 6). Cytotoxicity assay in human osteosarcoma cells confirms that the complex nanocarriers have shown no cytotoxicity by themselves but stimulate noteworthy cell growth inhibition while loading with doxorubicin, a chemotherapeutic drug. Under physiological pH, in vivo report showed no significant cargo release over 96 h. However, temporary exposure to acidic pH discharges an experimental fluorescent cargo for over 72 h. 

### 2.5. Capacitors

Capacitors are energy storage devices which deliver high power density as well as incredible energy. Recently, these features have offered great advantages as electric vehicles, electronic components, and backup power systems due to their energy storage benefits. These devices are based on an energy storage mechanism resulting from the synthesis of an electric double layer at the interface between an electrolyte solution and an electronically conductive material [24,163,164,165]. A classic supercapacitor consists of two carbon electrodes interposed by a porous matrix (separator) (Figure 1). The energy accumulated in carbon-based supercapacitors is a function of several parameters, such as pore structure and specific surface area of electrodes, voltage stability, and ionic conductivity of electrolyte. Moreover, the usual materials employed in traditional electrodes should have a high surface area, e.g., carbon aerogels [166,167], carbon fibers [168], and activated carbons [169,170,171]. The electrolytes commonly utilized in supercapacitors are an aqueous solution of H_2_SO_4_ or KOH [172] or non-aqueous solutions of tetraethyl-ammonium-tetrafluoroborate (ET_4_NBF_4_) in organic solvents [169,173]. The addition of solid polymer electrolytes in supercapacitors, as a liquid electrolytes substitute, may offer several benefits, including flexible structure, compact geometry, and easier packaging. Additionally, since this device is free from outflows of dangerous and corrosive liquids, it offers safety profiles. If mesoporous carbon with a well-ordered pore structure and a high specific surface area are used, the well-engineered solid-state supercapacitors would provide higher energy as well as power density compared to liquid electrolyte-based supercapacitors. Moreover, this capacitor would provide an advantage in terms of ease of design and realization of lightweight and flexible devices.

Recently, nanostructured mesoporous carbon materials achieved much interest as negative electrodes of rechargeable lithium batteries [174,175] and supercapacitor electrodes [171]. Tamai et al. [90] revealed that the introduction of mesoporous carbon in the electrodes improved the specific capacitance of carbon-based supercapacitors. Yamada et al. [91] comparatively studied various carbons of similar specific surface areas with various pore size distributions and found that highly mesoporous materials exhibited better capacitance ability. Yoon et al. [176] demonstrated that higher fractionated mesopores carbons delivered high power densities and exhibited a smaller time constant compared to microporous carbons. However, these experiments were based on liquid electrolyte supercapacitors [107] and the microstructural features of such carbon may affect the generation of effective polymer electrolyte-based supercapacitors.

Lufrano et al. [173] synthesized mesoporous carbons using SBA-15 silica templates and sucrose as a carbon source. These mesoporous carbon materials were used to develop composite electrodes by a casting method for solid-state supercapacitors. The electrochemical features of mesoporous carbons in supercapacitors were analyzed by electrochemical impedance spectroscopy and cyclic voltammetry. The CMK-3A mesoporous carbon showed a value of specific capacitance of 132 Fg^−1^ (for a single electrode), which was 68% higher than that of reference carbon material. The double-layer capacitor performance of the mesoporous carbon was 127% (12.05 µF cm^−2^ vs. 5.3 µF cm^−2^) when compared with the reference carbon. These findings were explained based on the high pseudocapacitance and other physicochemical properties of the mesoporous carbon, including accessibility of pores to the electrolyte, pore structure, and surface wettability by surface functional groups.

## 3. Future Challenges and Opportunities

Mesoporous carbon is regarded as the next generation inorganic material for many applications, including biomedical science, owing to its promising combinatorial properties, such as carbonaceous composition, distinct mesoporous structure, and high biocompatible nature [31]. However, in some applications, such as controlled drug release, mesoporous carbon materials are still at the developing stage. Additional improvements are necessary to boost the clinical application of this material: (a) scalable synthetic protocols with optimized structural as well as compositional parameters; to date, a very limited number of standard and controllable methodologies for the synthesis of mesoporous carbon have been reported, particularly for pore size-tunable and hydrophilic spherical material. (b) Surface modification of mesoporous carbon still remains challenging. The current oxidization scheme of mesoporous carbon might provide the carrier with many organic groups for additional modification; however, the strong oxidization partly affects the carbonaceous framework as well as the mesostructure of mesoporous material, leading to structural distortion and declined photothermal-conversion capability. More significantly, surface modification is crucial for well-ordered catalysts preparation, to supply drug carriers with smart drug delivery, and controlled and stimuli-responsive drug release, which has been hardly applied to mesoporous carbon, although extensively used in mesoporous silica. Compared to other materials, including graphene and mesoporous silica, mesoporous carbon has been rarely applied in the biomedical field and diagnostic imaging, although it has been expected to show good performance due to its unique structural composition and physicochemical features. For considering the use of mesoporous carbon in clinical translation in the upcoming future, it is urgently necessary to evaluate its biosafety, which intensely depends on the methodology of material synthesis. In addition, biosafety evaluation should be focused on some crucial parameters, including biodegradation, excretion, biodistribution, and other disease-specific toxicities, such as embryonic toxicity, reproductive toxicity, neurotoxicity, etc. An in vivo quantitative assay of mesoporous carbons might be challenging owing to their carbonaceous composition that may be affected by carbon present in the living system. In this case, radiolabeling of mesoporous carbon should be promising for biosafety evaluation.

Although the functional modification on mesoporous carbon surface gives excellent activity in various fields, it is still necessary to invent a cost-effective approach for large-scale applications with high stability as well as reliability. In addition to surface modification, the future challenge is to generate contamination-free functionalized material, since this material might be mixed up with the nanoporous material during synthesis. Functional materials with a high surface area were reported to be excellent materials for adsorption. Future challenges might arise to use the mesoporous material with both absorption and built-in electrochemical properties that may stimulate carbon material to be employed as electrocatalyst. Uncontrollable doping and irreversible agglomeration at the molecular level during high-temperature pyrolysis in their synthetic procedure are more challenging and need to be solved for better application; for example, when used as catalytic supports or as adsorbents. Furthermore, the lower synthesis of mesoporous carbon materials by the current methodology remains a problem from an industrial point of view, which should be taken into consideration during material synthesis.

## 4. Conclusions

In this review, the authors provide a general overview of the recent literature regarding various applications of mesoporous carbon. The limitation of the synthesis procedure of mesoporous carbon for potential applications is presented. Particular emphasis is provided to some areas, especially in surface modification, oxidation as well as functionalization time for suitable application. The role of reaction pH, reaction time, time of carbon-surface oxidation, functionalization parameters, solvent uses during adsorption, and optimal time in adsorption has been demonstrated for better understanding. Future research should focus more on the synthesis procedures, surface modification, and novel modification parameters for the promising application of mesoporous carbon.

## Figures and Tables

**Figure 1 ijms-22-04498-f001:**
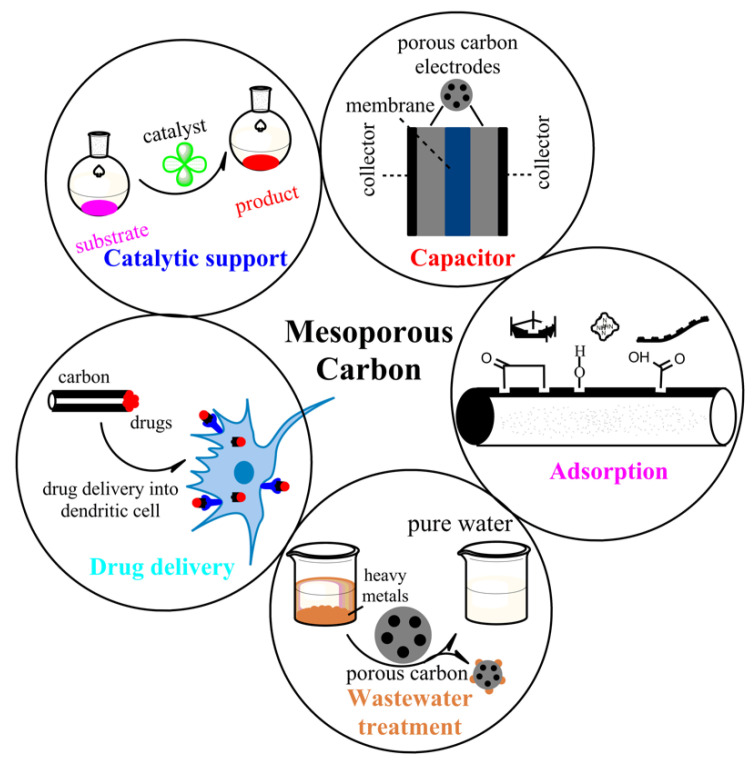
Potential applications of mesoporous carbon in modern scientific research.

**Figure 2 ijms-22-04498-f002:**
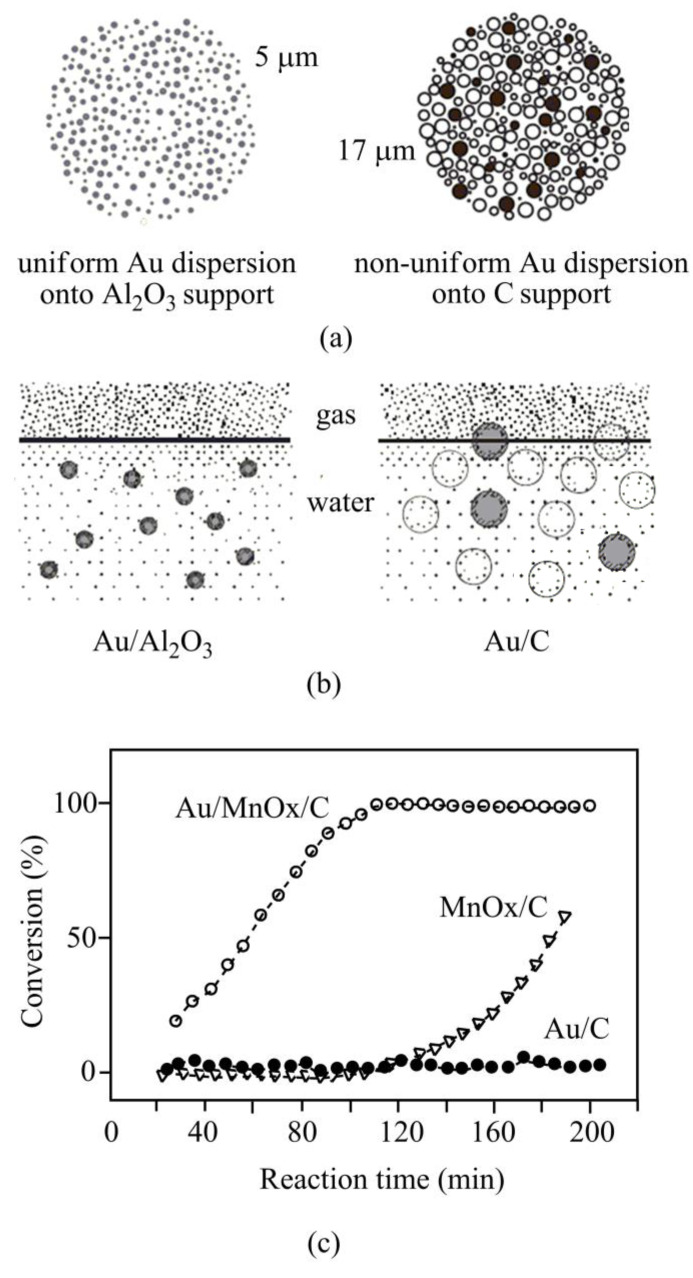
Application of carbon supports in various oxidation reactions. Comparative analyses on Au/Al_2_O_3_ and Au/C catalysts (**a**,**b**). Uniform gold distribution onto alumina supports at high glucose to Au ratios (**a**). Hydrophobic carbon supports greatly adhere to gas–liquid interface assisting oxygen mass transfer towards catalytic sites (**b**), adapted with permission from [44]. Effects of carbon functionalization in carbon-monoxide oxidation (**c**), adapted with permission from [51].

**Figure 3 ijms-22-04498-f003:**
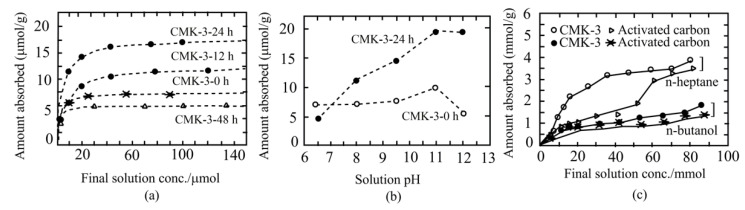
Adsorption of lysozyme and vitamin E over mesoporous carbon surfaces. Lysozyme adsorption dependency on the APS concentration and oxidation times, and pH of the solution (**a**,**b**). Comparative analyses of lysozyme adsorption on various forms of activated carbons (**a**) and dependency of pH on the adsorption properties of lysozyme enzyme (**b**), adapted with permission from [57]. Adsorption ability of vitamin E onto carbon surfaces from n-heptane and n-butanol at 293 K (**c**), adapted with permission from [58].

**Figure 4 ijms-22-04498-f004:**
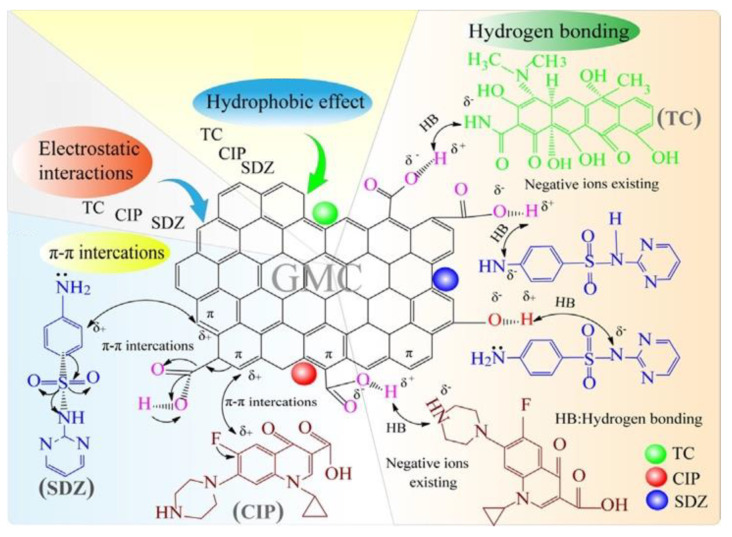
The mechanisms of the target antibiotics adsorption on GMC surface from aqueous solutions. Adapted with permission from [62].

**Figure 5 ijms-22-04498-f005:**
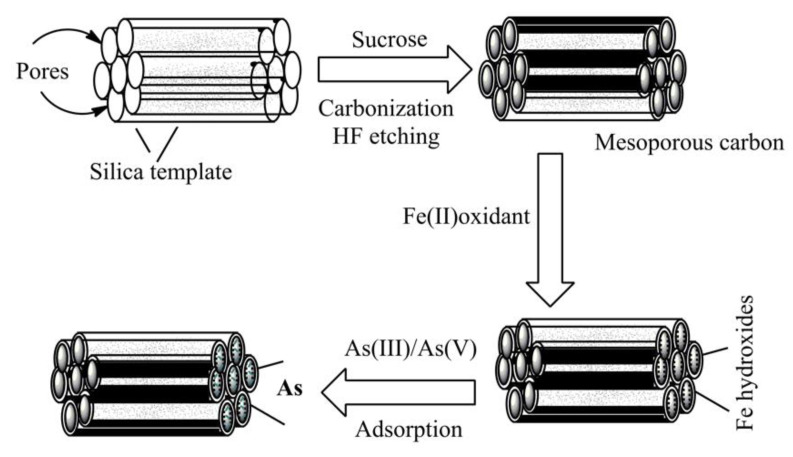
A graphical model for the preparation of carbon-based adsorbents and their application in arsenic adsorption. Silica has been used as the template for the synthesis of mesoporous carbon by carbonization followed by iron coating for the removal of metal ions from drinking water.

**Figure 6 ijms-22-04498-f006:**
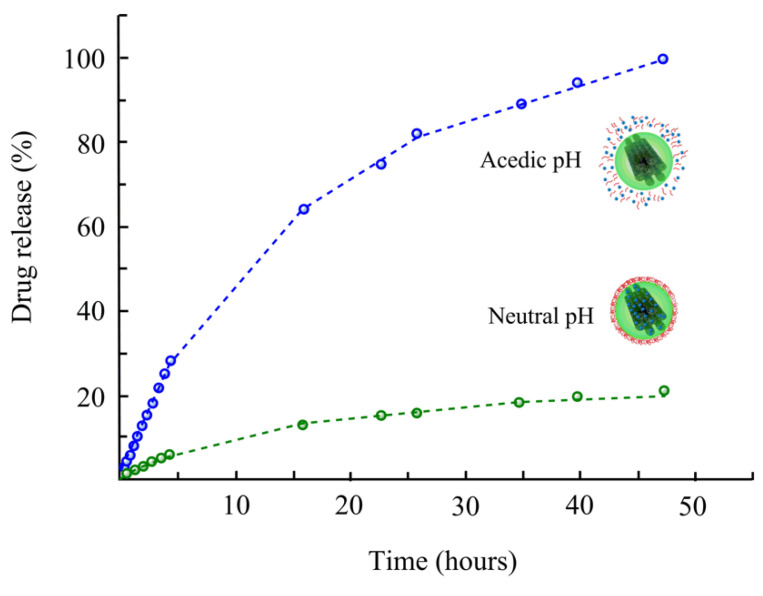
Schematic diagram of the pH-responsive mesoporous carbons. At physiological pH, the self-immolative coating remains collapsed on the surface. Conversely, at acidic pH, the polymers undergo self-immolation, leading to cargo release. Adapted with permission from [84].

**Table 1 ijms-22-04498-t001:** Application of mesoporous carbon in various scientific disciplines.

Disciplines	Form of Mesoporous Carbon	Preparation Method	Application	Ref.
Catalytic supports	Au/C	Deposition–precipitation, cationic adsorption	Glucose oxidation	[44]
	Au/C	Incipient wetness impregnation	Glucose oxidation	[45]
	Au–Pd/C	Impregnation	Glyoxal and glucose oxidation	[46]
	Au/C	Immobilization	Glucose and Alcohol oxidation	[47]
	OMCs/tungsten carbide composites	Soft template	Methanol electrooxidation	[48]
	Pt-Ru/C	Co-precipitation	Methanol electrooxidation	[49]
	Pt/C and PtCO_3_O_4_/C	Microwave and Impregnation	Methanol oxidation	[50]
	Au/MnO_x_/C	Electrodeposition	CO oxidation	[51]
	Au/TiO2/C	Sonochemical and Microwave	CO oxidation	[52]
	Porous carbon supported gold catalysts	Antigalvanic reduction	Oxygen reduction reaction	[53]
	Mesoporous carbon supported gold	Hydrothermal synthesis	Reduction of nitroarenes	[54]
	Ordered mesoporous carbon supported gold	Wet chemical	Oxygen reduction	[55]
	Au/FeO_x_	Deposition	CO oxidation	[56]
Adsorbents	CMK-3–100 (3 nm) and CMK-3-150 (6.5 nm)	Template synthesis	Lysozyme adsorption	[57]
	CMK-1 and CMK-3	Template synthesis	Vitamin E adsorption	[58]
	Activated carbon with mesopores	Commercial	Sugars adsorption	[59]
	Activated carbon with mesopores	Commercial	Sugars adsorption	[60]
	Mesoporous carbon	Template synthesis	Antibiotics adsorption	[61]
	Glucose-based mesoporous carbon	Template synthesis	Antibiotics adsorption	[62]
	Activated carbon with mesopores	Commercial	Sugars adsorption	[63]
Wastewater treatment	Magnetic mesoporous carbon	Wet impregnation	Removal of methyl orange and methyl blue	[64]
	Iron containing mesoporous carbon	Template synthesis	Removal of Arsenic (As)	[65]
	Magnetically graphitic mesoporous carbon	Template synthesis	Removal of Chinese medical waste	[66]
	Magnetically encapsulated mesoporous carbon	Template synthesis	Removal of methylene blue, Congo red	[67]
	Ordered mesoporous carbon	Template synthesis	Removal of malachite green	[68]
	Boron-doped mesoporous carbon nitride	Template synthesis	Removal of malachite green	[69]
	Mesoporous carbon nanofibers	Hydrothermal	Removal of methylene blue, methyl orange	[70]
	S-doped magnetic mesoporous carbon	Template synthesis	Removal of methyl orange	[71]
	Magnetic mesoporous carbon nanospheres	Template synthesis	Removal of hexavalent chromium	[72]
	Polyacrylic acid modified magnetic mesoporous carbon	Template synthesis, co-impregnation	Removal of Cd(II)	[73]
	Ordered mesoporous carbon electrodes	Template synthesis	Copper (II)	[74]
	Boron doped ordered mesoporous carbon	Template synthesis	Pb(II)	[75]
	Functionalized mesoporous carbon	Chemical modification	Pb(II)	[76]
	Phosphate modified ordered mesoporous carbon	Template synthesis	Pb(II)	[77]
	Ordered mesoporous carbon	Template synthesis	AV90 dye	[78]
	Modified mesoporous carbon	Template synthesis	Bisphenol-A	[79]
	Octyl modified ordered mesoporous carbon	Template synthesis	Phenol	[80]
	Functional mesoporous carbon	Hydrothermal carbonization	Bisphenol and diuron	[81]
	Mesoporous carbon microsphere	Template synthesis	Removal of hexavalent chromium	[82]
Drug delivery	Mesoporous carbon	Template synthesis	Celecoxib	[83]
	Mesoporous carbon nanoparticles	Template synthesis	Ruthenium dye	[84]
	ZnO gated mesoporous carbon nanoparticles	Template synthesis	Mitoxantrone	[85]
	CMK-1 type mesoporous carbon nanoparticle	Template synthesis	Fura-2	[86]
	Folate functionalized mesoporous carbon	Template synthesis	Doxorubicin	[87]
	Hyaluronic acid modified mesoporous carbon nanoparticles	Template synthesis	Doxorubicin	[88]
	Mesoporous carbon nanospheres	Hydrothermal synthesis	Doxorubicin	[89]
Capacitors	Mesoporous carbon	Carbonization	Electric double layer capacitors	[90]
	Mesoporous carbon	Defluorination	Electric double layer capacitors	[91]

## Data Availability

Not applicable.

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
