# Peer review of "Mesoporous Carbon: A Versatile Material for Scientific Applications"

_ijms, 2021, doi:10.3390/ijms22094498_

Round 1

Reviewer 1 Report

The manuscript entitled “Mesoporous carbon: a versatile material for scientific applications– a Review” presents the general overview the recent literature concerning the many applications of mesoporous carbon materials. The authors focused on applications in following areas: catalytic supports, adsorption, waste-water treatment, drug delivery and energy storage devices.

The role of various parameters influencing the synthesis of mesoporous carbons was also shown: reaction pH and time, time of carbon-surface oxidation, functionalization parameters, solvent uses during adsorption, and optimal time in adsorption. This work presents information on the basis of 159 references, of which 18 references are from 2019-2020, so this manuscript may be a useful source of knowledge for other researchers. The lower value of this review is affected by the fact that similar reviews have already been published, e.g.

Mercy R. Benzigar, Siddulu Naidu Talapaneni,   Stalin Joseph, Kavitha Ramadass,   Gurwinder Singh,   Jessica Scaranto,   Ugo Ravon, Khalid Al-Bahily, Ajayan Vinu. Recent advances in functionalized micro and mesoporous carbon materials: synthesis and applications. Chem. Soc. Rev., 2018,47, 2680-2721

Wang Xin, Yonghui Song. Mesoporous carbons: recent advances in synthesis and typical applications. RSC Adv., 2015,5, 83239-83285

I recommend this manuscript for publication.

Author Response

Response: Thank you for your appreciation, comments and time to review the manuscript.

Reviewer 2 Report

From my point of view, the publication of the ‘’review article (Mesoporous carbon: a versatile material for scientific applications) can only be considered after major revisions.

In the introduction, the part should be improved and the author would be discussed more about mesoporous carbon. The author should explain in detail the motivation of this review in the introduction part.

 Re-draw Figure 2 especially (b)

Add the “Future Challenges and opportunities” section before the conclusion.

Please check language, typos, a chemical formulas such as page 6 line 114, page 7 line 152.

Author Response

Response to Reviewer 2 Comments

Point 1: From my point of view, the publication of the ‘’review article (Mesoporous carbon: a versatile material for scientific applications) can only be considered after major revisions.

Response 1: Thank you for the comments.

Point 2: In the introduction, the part should be improved and the author would be discussed more about mesoporous carbon.

Response 2: The authors appreciate your comments. We added a paragraph regarding the mesoporous carbon to the introduction section and marked it with yellow color

Point 3: The author should explain in detail the motivation of this review in the introduction part.

Response 3: Thank you for your suggestion. We added a section regarding the motivation at the bottom of the introduction part and marked it with yellow color

Point 4: Re-draw Figure 2 especially (b)

Response 4: Thank you. We understand the point. Figure 2b has been replaced.

Point 5: Add the “Future Challenges and opportunities” section before the conclusion.

Response 5: Thank you. According to the reviewer’s suggestion, Future Challenges and opportunities have been added to the revised version just before conclusion and marked with yellow.

Point 6: Please check language, typos, a chemical formula such as page 6 line 114, page 7 line 152.

Response 6: These issues have been resolved and marked with yellow color.

In addition to these, the authors checked the grammar moderately and rearranged the references, and all the changes have been marked with yellow.

Round 2

Reviewer 2 Report

I believe the manuscript has been significantly improved and now warrants publication in IJMS.